# Nitrogen Self-Doped Metal Free Catalysts Derived from Chitin via One Step Method for Efficient Electrocatalytic $CO_2$ Reduction to CO

Peixu Sun [1,2], Xiaoxiao Wang [1,2], Mingjian Zhu [1,2], Naveed Ahmad [1,2], Kai Zhang [1,2] and Xia Xu [1,2,*]

[1] School of Chemistry and Chemical Engineering, Anhui University of Technology, Ma'anshan 243002, China
[2] Biochemical Engineering Research Center, Anhui University of Technology, Ma'anshan 243002, China
* Correspondence: xiax@hotmail.com

**Abstract:** In this study, a facile one-step method via pyrolysis was used to prepare nitrogen self-doped metal free catalysts derived from inexpensive biomass-chitin for an electrochemical $CO_2$ reduction reaction ($CO_2$RR). The microstructure, surface area, defect and N type in the catalysts were analyzed by BET, Raman, XPS, SEM and TEM. The sustainable chitin-based electrocatalyst prepared under optimized conditions has a surface area of 1972 $m^2/g$ and can convert $CO_2$ into CO with $FE_{CO}$ of ~90% at a potential of $-0.59$ V (vs. RHE). This good $CO_2$RR performance results from plentiful active sites due to a high surface area, rich ultra-micropores that are beneficial to $CO_2$ adsorption, abundant mesopores for $CO_2$ transport improvement, a high content of pyridinic and graphitic nitrogen that is favorable for a $CO_2$ reduction reaction and a low interfacial charge transfer resistance leading to a rapid electron transfer rate from the catalyst to $CO_2$. This study shows the feasibility of N self-doped biomass-derived catalysts for $CO_2$RR with the potential for large-scale industrial applications.

**Keywords:** biomass; $CO_2$ electroreduction; nitrogen self-doped catalyst; metal free catalysts

## 1. Introduction

In recent years, the continuously rising level of $CO_2$ in the atmosphere caused by the excessive use of fossil fuels has become the primary drive of environmental problems, leading to serious impacts on sustainable development [1]. To reduce $CO_2$ emission, technology for efficient $CO_2$ capture and conversion is highly demanded. Direct conversion of $CO_2$ to valuable products has become of considerable interest [2,3]. Among various approaches, an electrochemical $CO_2$ reduction reaction ($CO_2$RR) coupling with a renewable energy source [4] is a promising method for achieving a carbon-neutral energy cycle and conversion to hydrocarbons at mild operation conditions [5–7].

Due to the thermodynamic stability of $CO_2$, high overpotentials are always required to convert $CO_2$ into the desired chemicals and fuels [8–12]. Among the various reduction processes, the reduction of $CO_2$ to CO, which has a relatively high selectivity and a facile separation procedure, is one of the most promising practices. Over the past decades, commonly used electrocatalysts for CO evolution are metal-based catalysts such as noble metal-based catalysts (Au, Ag) [13,14] and transition metal-based catalysts (Ni, Fe) [15,16]. Although metal-based catalysts for $CO_2$RR have several weaknesses, such as high cost, poor stability and non-renewability [17,18], metal-free-based catalysts are much more attractive for electrochemical $CO_2$ reduction.

Due to intrinsic advantages such as low cost, natural abundance and excellent thermal/chemical stability, metal free-based electrocatalysts provide a promising platform for converting $CO_2$ to CO [19–21]. Carbon-based electrocatalysts, a type of metal free catalyst, have emerged as alternative catalysts to be used for $CO_2$RR [22]. Massive efforts have been made to construct carbon-based metal-free catalysts by several strategies, such as atomic doping and defecting of carbon lattice [23], to improve catalytic activity of pristine

graphene for $CO_2RR$ [24]. All these approaches create active sites in metal-free-based catalysts, leading to changes in electronic structures and modulating the adsorption energy and reaction activation energy of intermediates for $CO_2$ reduction [25,26]. An abundance of pyridinic-N was demonstrated to be an active site for high $FE_{CO}$ of 92% and 84% on the wrinkle-free and wrinkled carbon nanosheets [27,28] and 85 and 89% on the graphene foam and graphene nano-ribbon network [29,30]. Defect-rich carbon catalysts prepared by N removal are around two-fold greater than edge-rich graphene [31], with $FE_{CO}$ of around 80% at $-0.6$ V [32]. Hence, the combination of atomic doping with the defect engineering is a favorable strategy for enhancing CO's selective performance in $CO_2RR$.

Biomass is a renewable precursor with great potential for synthesizing carbon-based electrocatalysts for diverse electrocatalytic reactions [33]. However, the direct conversion of biomass into a highly active electrocatalysts for $CO_2RR$ is reported only in limited studies. Wood- and wheat-derived carbons with N-doping assist the conversion of $CO_2$ into CO [34,35]. Hao et al. developed a cedar biomass-derived N-doped graphitized carbon electrocatalyst via a two-step preparation, achieving an $FE_{CO}$ of 91% at $-0.56$ V (vs. RHE) and a partial current density (PCD) to CO at about 3.7 mA cm$^{-2}$ [36]. A defect-rich carbon electrocatalyst derived from the nitrogen-rich silk cocoon prepared using two-step carbonization catalyzed $CO_2$ reduction to CO with an $FE_{CO}$ of ~89% at $-1.09$ V (vs. RHE) but PCD of CO at around 1 mA cm$^{-2}$ [37].

N abundant biomass as a potential carbon source is an ideal source for preparing carbon-based electrocatalysts. Due to the efficiency of the extraction process, chitin from shrimp, crab, crayfish and krill shells (around $6.8 \times 10^6$ ton/year) is considered to be a sustainable source for removing $CO_2$ or $CO_2$ conversion, although the sustainability of chitin production should be evaluated in a holistic way, considering the life cycle assessment of the whole system. Chitin-based N-doped carbon was prepared as bifunctional catalysts for an oxygen reduction reaction (ORR) [38]. A sponge-like three-dimensional (3D) network architecture derived from chitin via low-temperature dissolution and subsequent carbonization was obtained with many active sites to $O_2$, exhibiting pronounced ORR and oxygen evolution reaction (OER) activity [39]. From a sustainable point of view, catalysts derived from renewable carbon sources with simple and easy synthesis routes are essential and highly needed.

To the best of our knowledge, no catalysts derived from chitin for $CO_2RR$ have been explored. In this study, chitin was used as both carbon and N precursors to prepare metal-free catalysts for $CO_2RR$ using a facile synthesis procedure. The $CO_2RR$ performance was determined for catalysts prepared using different activators under pyrolysis at different temperatures. The microstructure, surface area, defect, N type and content in the catalysts were analyzed using BET, Raman, XPS, SEM and TEM. The electrocatalyst synthesized under optimized conditions exhibits a good $CO_2RR$ performance. These results imply the feasibility of using nitrogen self-doped chitin-derived catalysts with high selectivity.

## 2. Results and Discussion

### 2.1. Preparation and General Characterization of Chitin-Derived Catalysts

Due to high crystallinity and stability, it is difficult to use chitin directly. Strong Lewis acids such as $FeCl_3$ and $ZnCl_2$ are usually used as activators to produce a high surface area, porous structure and intrinsic defects in biomass-derived catalysts [40]. Here, the chitin was initially pyrolyzed at 400 °C for 1 h and then 1 h at 900 °C at different ratios of chitin to $FeCl_3$ (1:2, 1:2.5, 1:3 and 1:4). TGA was applied to investigate the weight loss during carbonization in the presence of $FeCl_3$ (Figure S1) to understand the carbonization and activation processes. Chitins with $FeCl_3$ exhibited two sharp drops at round 200 °C and 750 °C while a dramatic decline in the weight loss at around 400 °C was observed in the absence of $FeCl_3$. These suggested that $FeCl_3$ initially reacted with $H_2O$ to form FeOCl (Equation (S1)) and then $Fe_2O_3$ (Equation (S2)) with HCl release, corresponding to a sharper drop at around 200 °C, leading to the reduction in the crystallinity and stability of chitin. Then, as the temperature increased, the carbonization continued with the greatest

mass loss rate while $Fe_2O_3$ gradually reduced to $Fe_3O_4$, FeO and Fe in step with the CO and $CO_2$ release (Equations (S3)–(S7)).

The bands at 3420, 3255 and 3097 $cm^{-1}$ corresponded to an intramolecular hydrogen bond, the intermolecular hydrogen bond CO–NH and the intermolecular NH of chitin, respectively, as shown in Figure 1a. The vibration at ~2900 $cm^{-1}$ was attributed to the CH vibration, the peaks at 1650 $cm^{-1}$ and 1550 $cm^{-1}$ showed the presence of amide I and amide II, respectively, and the peaks at 1116 and 1025 $cm^{-1}$ were assigned to the stretching vibrations of C–O–C [41]. The 1650 $cm^{-1}$, 1550 $cm^{-1}$ and 1116 $cm^{-1}$ peaks disappeared after carbonization, indicating that the bonds in amide I and II and C–O–C were destroyed. The new peaks at 1600 $cm^{-1}$ and 805 $cm^{-1}$ for the activated catalysts became stronger with the ratio of Fe to chitin increasing, implying the formation of defect sites [42–44].

The defects and graphitization degrees of the obtained catalysts were then characterized. As shown in Figure 1b, two broad bands at ~1350 $cm^{-1}$ and 1600 $cm^{-1}$ were attributed to D-band for disordered carbon atoms and defects and G-band for an ordered graphitic vibration, respectively. The presence of $FeCl_3$ during pyrolysis dramatically increased the $I_D/I_G$ (G bands represent the graphite-type lattice vibrations and D bands reflect the disordered graphite lattice vibrations) values of chitin-derived carbon materials, indicating that the abundance of intrinsic defects in carbon-based catalysts [37,45,46] may arise from nonplanar microstructure distortions or lattice defects. A slightly higher $I_D/I_G$ value suggested more intrinsic defects in chitin-2.5Fe that probably included lattice vacancies and edge dislocations. Consistent with the Raman results, chitin-0Fe exhibited a sharp peak at $2\theta$ = ~26.9° that was associated with a characteristic peak of graphite and a broad peak centered at $2\theta$ = ~43° (Figure 1c) that corresponded with a typical amorphous carbon structure in X-ray diffraction patterns [47]. By contrast, the presence of activator $FeCl_3$ led to a decrease in the graphite and resulted in more amorphous carbon structures, as indicated by two broad peaks at around 26.9° and 43°. In addition, no sharp peak at 44.9° attributed to the iron plane (110) [48] was found.

The N, C, O and Fe contents in the prepared catalysts were determined using XPS. No Fe element was observed (Figure 1d), indicating that no doped Fe existed in the prepared catalysts. All the XPS and XRD results together indicated that no Fe element was present in the chitin-derived catalysts. The deconvolution of XPS spectra was performed to differentiate the N types present in the chitin-derived catalysts (Figure 1e). The XPS results illustrated four different nitrogen peaks of pyridinic nitrogen at 398.3 eV, pyrrolic nitrogen at 400.1 eV, graphitic nitrogen at 401.5 eV and oxidized nitrogen at 402.1 eV, which can affect the charge of N atoms and C atoms [32,33]. The increasing amount of $FeCl_3$ during pyrolysis led to an elevation in the percentage of total N content in the chitin-derived catalysts, which reached its maximum at the ratio of 1:3. The total percentage of pyridinic and graphitic N reached the maximum at 1:2.5 (Figure 1f), but the percentage of pyridinic N reached the maximum at the ratio of 1:2.5. The total percentage of pyridinic and pyrrolic N, which only existed in the zigzag edge or the defect sites, increased due to the introduction of $FeCl_3$, suggesting more defect sites, consistent with the Raman results (Figure 1b). In addition, the C/O and C/N atomic ratio reduced, but no further decrease was observed when the chitin/$FeCl_3$ ratio was $\geq$1:2.5 (Figures S2 and S3).

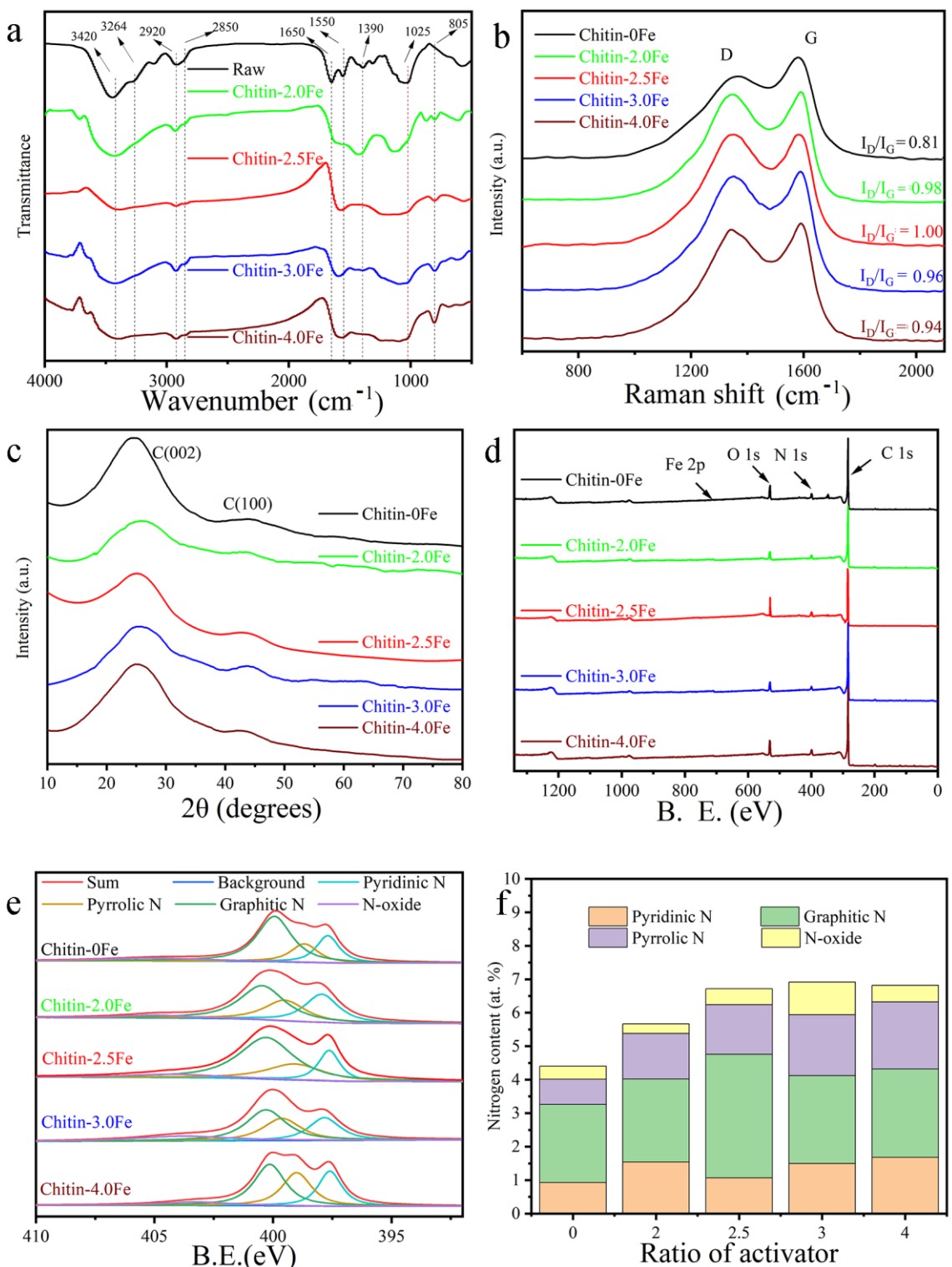

**Figure 1.** Characterization of chitin-derived catalysts prepared with different ratios of chitin to FeCl$_3$. (**a**). FTIR, (**b**). Raman, (**c**). XRD, (**d**,**e**). XPS, (**f**). N content.

### 2.2. Morphology and Pore Characterization

Due to the removal of light and volatile matters during pyrolysis, SEM was employed to investigate the morphology and textural characteristics' change after carbonization (Figure 2). The chitin-0Fe exhibited dense structure with a smooth surface. Gas generation during carbonization in the presence of FeCl$_3$ manipulated the structures of the pores

produced, leading to the hierarchical porous structure, which is beneficial for exposing more active sites as well as electrolyte diffusion. However, the pores appeared to be relatively less interconnected when the FeCl₃ amount was increased from 1:2.5 to 1:4. All taken together, a 1:2.5 weight ratio of chitin to FeCl₃ was deemed to be the optimal ratio. The chitin-derived catalysts with FeCl₃ exhibited nanosheet-like morphology with wrinkles and a 3D cross-linked network structure. During the catalyst preparation, HCl was used to thoroughly wash out the Fe residues to obtain metal-free biomass-derived catalysts according to the previous studies [36,37]. However, the SEM images showed dark particles not only in chitin-2.0Fe, chitin-2.5Fe, chitin-3.5Fe and chitin-4.0Fe but also in chitin-0Fe, suggesting that the dark particles could be caused by deposition, further indicating no Fe element in the chitin-derived catalysts, as is consistent with the XPS and XRD results.

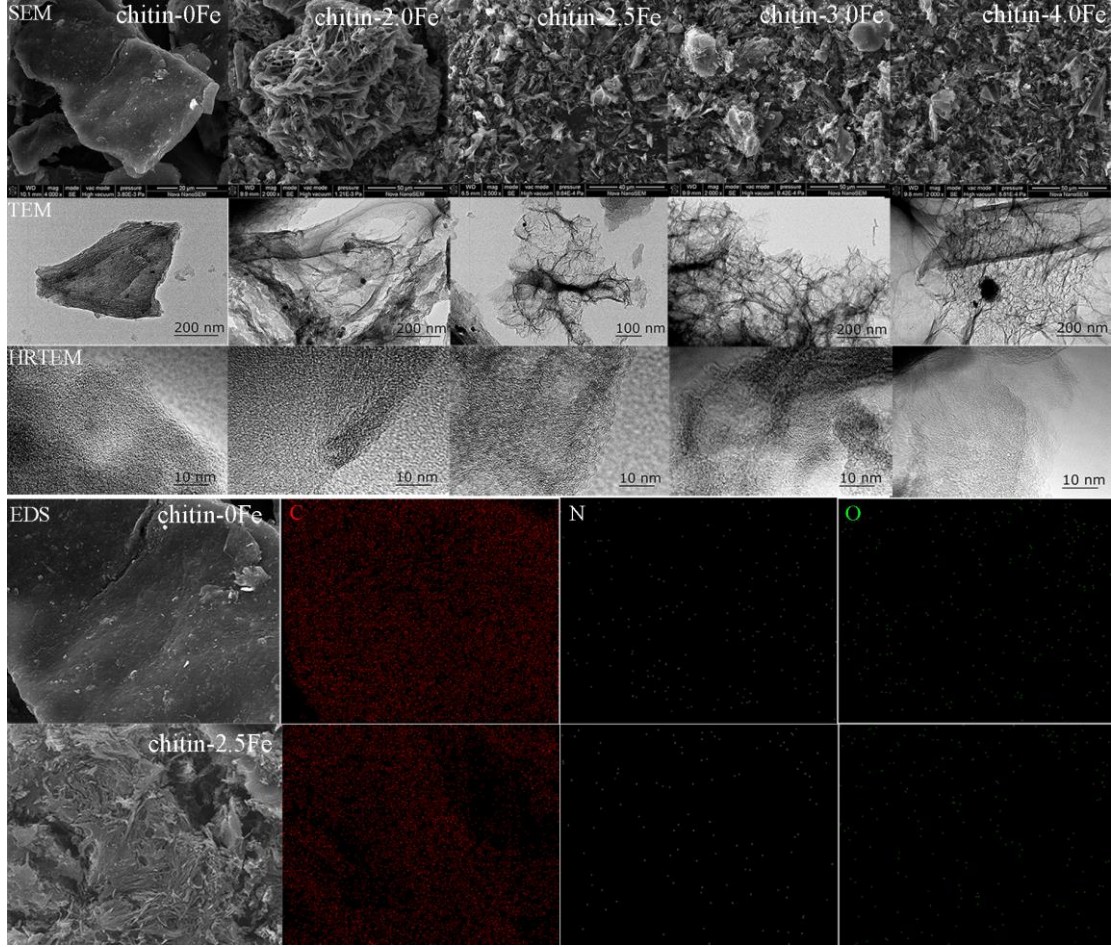

**Figure 2.** Morphological characterization for prepared catalysts obtained at different ratios of chitin to FeCl₃.

The FeCl₃ activation led to a more amorphous structure, as is consistent with the Raman results (Figure 1c). A distorted graphitic lattice in the enlarged images in HR-TEM images was observed for the chitin-derived catalysts with FeCl₃, suggesting more active sites, whereas the chitin-0Fe exhibited a clear (0 0 2) plane of graphitic lattice and a smooth surface, as is consistent with the XRD analysis results (Figure 1c). The presence of FeCl₃ during pyrolysis clearly led to the amorphous structure, which is in line with the silk-derived catalysts prepared via pre-carbonization and activation using ZnCl₂ [37] but not consistent with the N-doped carbon prepared from cedar with melamine [36]. The EDS results revealed that nitrogen, oxygen and carbon were evenly distributed.

The nitrogen adsorption/desorption isotherms, pore size distributions, incremental pore volume, cumulative pore volume and cumulative surface area are shown in

Figures 3 and S4, respectively. All the samples exhibited a typical type-I curve except for the chitin-2.5Fe (Figure 3a). $FeCl_3$ improved the specific surface area and porosity via synergistic effects of carbon lattice expansion and chemical activation (Figure 3b–d). A sharp rise in the nitrogen adsorption at a low relative pressure ($P/P_0 < 0.01$) indicated a high concentration of micropores in all of the samples. At a ratio of 1:2.5, an initial sharp rise in nitrogen uptake was observed, while a further substantial increase suggested a highly microporous and mesoporous structure owing to the $Fe_2O_3$ decomposition and CO and $CO_2$ release (Equations (S3)–(S7)). Unlike the chitin-0Fe, the cumulative surface area and pore volume of other samples increased rapidly in the pore size region ranging from 0.7 nm to 1.1 nm; a further sharp increase was observed for the chitin-2.5Fe in the pore size region ranging from 1.1 nm to 3 nm in the somewhat mesoporous region. Moreover, in comparison with other conditions, the chitin-2.5Fe exhibited similar number of micro-pores (<1 nm), but remarkably more micro-pores (1–2 nm) and meso-pores (2–3 nm), showing the greatest surface volume and pore volume. An increasing slope at a relative pressure from 0.1 to 0.4 in the chitin-2.5Fe further suggested that the pores were wider, confirming the existence of meso-pores in the chitin-2.5Fe, implicating the improved mass transfer.

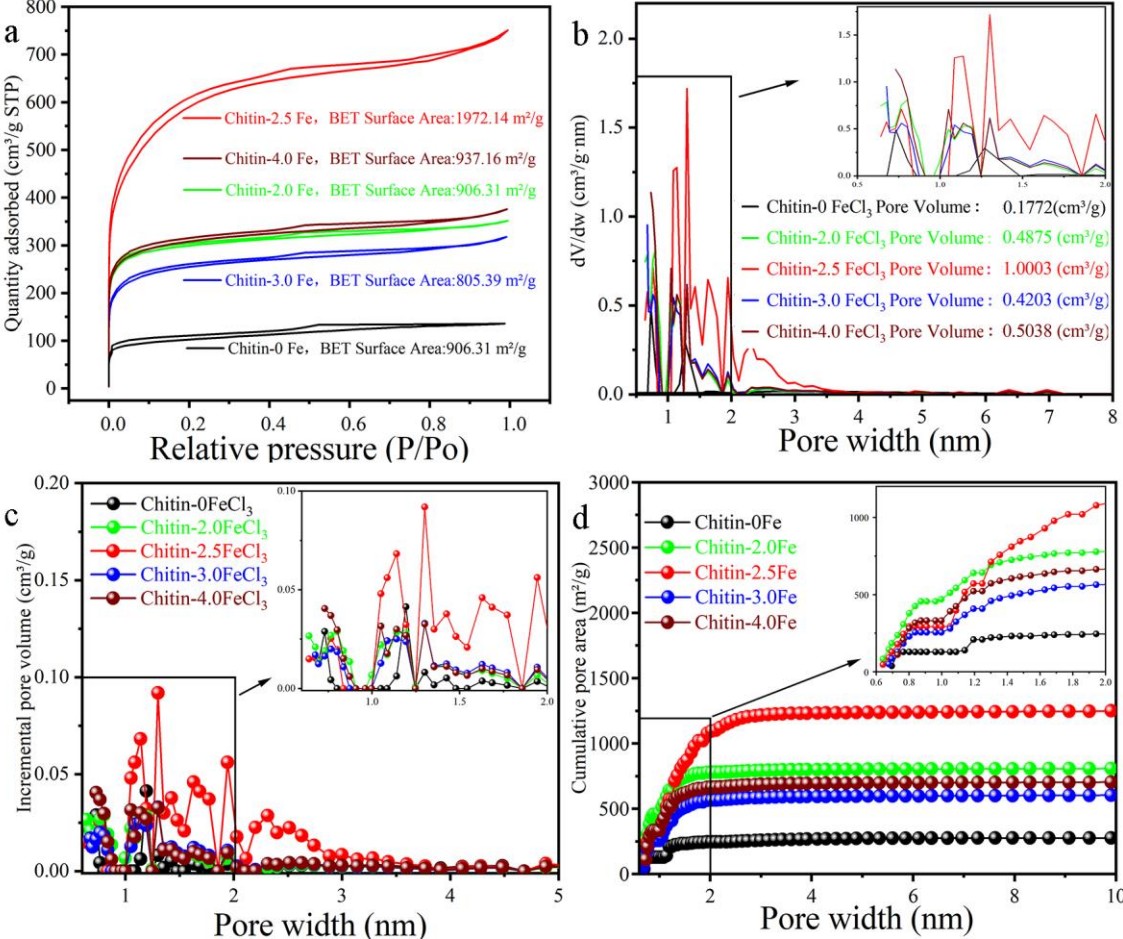

**Figure 3.** Pore characterization of the catalysts prepared at various ratios of chitin to $FeCl_3$. (**a**). $N_2$ absorption/desorption at 77 K, (**b**). Pore distribution, (**c**). Incremental pore volume, (**d**). Cumulative pore area.

### 2.3. CO_2RR Performance

The $CO_2RR$ performance was tested in an aqueous solution of 0.1 M $KHCO_3$ with $CO_2$ saturation. According to LSV and CV studies (Figure 4a,b), all the samples prepared with $FeCl_3$ possessed a better ability of $CO_2RR$ than those prepared without $FeCl_3$. The chitin-2.5Fe electrocatalysts exhibited a low onset potential and relatively higher reducing current

than the others. The catalytic performance was evaluated under the constant potentials 0.39, −0.49, 0.59 and −0.69 V (vs. RHE). Since the summation of FE related to $H_2$ and CO was nearly 100%, indicating that the gaseous products only included CO and $H_2$ and no liquid products, it was found that $FeCl_3$-activated catalysts enhanced the selectivity to CO at all tested potentials whereas chitin-0Fe displayed negligible activity for conversing $CO_2$ to CO, which was perhaps mainly caused by nonporous morphology.

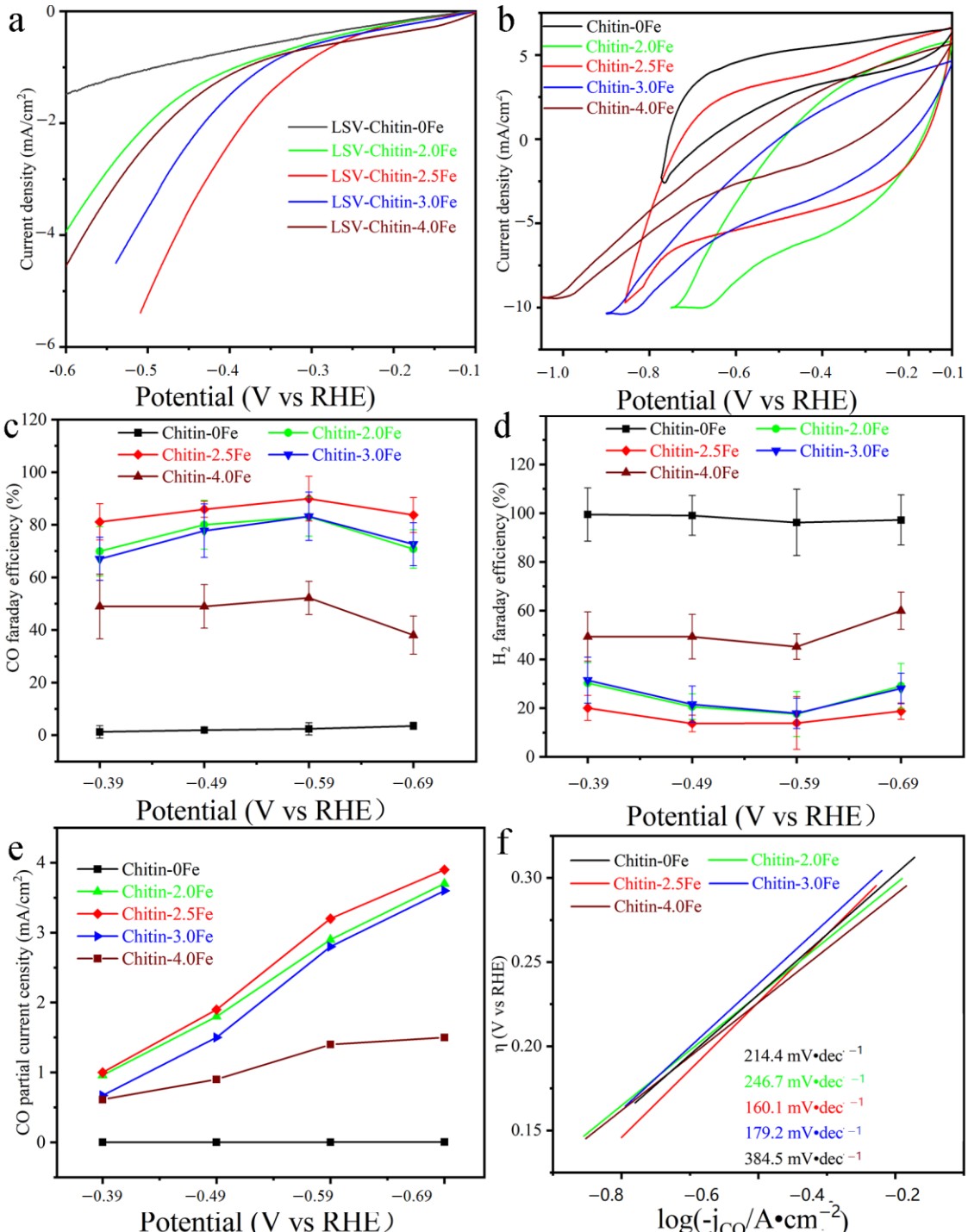

**Figure 4.** Electrocatalytic $CO_2RR$ performance over the chitin-derived catalysts prepared at different weight ratios of chitin to $FeCl_3$. (**a**). LSV, (**b**). CV, (**c**–**e**). $FE_{CO}$ and $FE_{H2}$ and PCD for CO, (**f**). Tafel slope.

The chitin-2.5Fe exhibited higher $FE_{CO}$ and less $FE_{H2}$ at each applied potential than the others, reaching up to 80% of $FE_{CO}$ even at an applied potential as low as $-0.39$ V (vs. RHE) and showing the highest $FE_{CO}$ of around 90% at $-0.59$ V (vs. RHE) and strongest suppression of $H_2$ evolution (Figure 4c,d). We also determined $FE_{CO}$ and $FE_{H2}$ of a carbon plate (Figure S7). Almost no CO was detected, indicating that the coating of chitin-derived led to the enhancement of $FE_{CO}$. Figure 4e shows that the partial current density ($j_{CO}$) increased with the applied potentials from $-0.39$ to $-0.69$ V (vs. RHE) and followed the order chitin-2.5Fe > chitin-2.0Fe > chitin-3.0Fe > chitin-4.0Fe > chitin-0Fe. The chitin-2.5Fe possessed a maximum value at each potential, showing the highest $j_{CO}$ of $\sim$3.5 mA/cm$^2$ at $-0.69$ V (vs. RHE). The different trend between $FE_{CO}$ and the percentage of pyridinic N in the catalyst suggested that the content of pyridinic N is not the only factor affecting the $CO_2$RR performance. Compared with the other catalysts, the best performance by chitin-2.5Fe could result from the highest specific surface area, pore volume and the total content of pyridinic and graphitic nitrogen. As seen in Figure 1f, the chitin-2.5Fe contained much less pyridinic nitrogen, which could by a reason for no dramatically superior selectivity to CO over the chitin-2.0Fe and chitin-3.0Fe.

The Tafel slope was calculated to be 160.10 mV dec$^{-1}$ for the chitin-2.5Fe (Figure 4f), suggesting that it offers more favorable kinetics for CO generation than the others. Considering the high $CO_2$RR selectivity and high surface area, which is beneficial for exposing active sites on the surface of the catalyst during the $CO_2$ electroreduction process, the ratio of chitin to FeCl$_3$ at 1:2.5 was regarded as the best ratio for further preparing the chitin-derived electrocatalysts for $CO_2$RR in this study.

### 2.4. $CO_2$RR Performance over Chitin-Derived Catalysts Prepared at Different Temperatures

To investigate the pyrolysis temperature effect on the $CO_2$RR performance, a series of catalysts was prepared at the temperatures of 700–1000 °C with chitin: FeCl$_3$ at 1:2.5, denoted as chitin-700 °C, chitin-800 °C, chitin-900 °C and chitin-1000 °C. They exhibited porous structure and nanosheet morphology with more defects (Figures 5a,b and S5). As seen in Figure 5c, the graphitic N content increased with the temperature until 900 °C, then a further rise in the calcination temperature resulted in diminishing the existing nitrogen functional groups (Figure 5c). The change trend in the pyridine and pyrrolic nitrogen content exhibited an inverse V shape (Figure 5c) while the C=O content displayed a V shape and the C–O content showed an S shape (Figure S6). PCD and maximum FE for CO over the chitin-700 °C, -800 °C and -900 °C showed a positive correlation with the graphitic N content and a negative correlation with the C=O content (Figures 5c and S6) but no direct correlation with the pyridinic N, implying that the pyridinic nitrogen defect sites are not the only factor for contribution to $FE_{CO}$ [36]. Despite higher nitrogen content in the chitin-800, it exhibited the similar $FE_{CO}$ and $FE_{H2}$ as the chintin-1000 °C (Figures 5d and S7), suggesting that the positive effect of high nitrogen content was offset by its low surface area and pore volume (Figure 5e). Due to the local electronic redistribution caused by the presence of nitrogen and oxygen functional groups, the carbon sites in the chitin-800 °C possessed larger charge densities, further leading to higher PCD than the chitin-1000 °C (Figure 5c).

In comparison with the reported metal-free carbon nanotube-based electrocatalysts (Table 1), the chitin-900 °C prepared in this work exhibited higher $FE_{CO}$ than MWCNT/Cc and g-C$_3$N$_4$/MWCNT [49,50], showed comparable $FE_{CO}$ to NCNTs-CAN-850 and N-CNT but higher $j_{CO}$ [51,52] and exhibited similar $FE_{CO}$ to NCNT-3-700 [53] but at lower overpotential. The chitin-900 manifested parallel $FE_{CO}$ to porous nitrogen-doped carbon catalysts with sulfur/fluorine/phosphorous addition but higher $j_{CO}$ at a similar overpotential [54–58] and displayed much better $CO_2$RR performance than the previous studies [34,59,60]. Compared with the N-doped electrocatalysts derived from biomass, the chitin-900 °C prepared using the simpler procedure showed high specific surface area [35–37] and better $FE_{CO}$ [35].

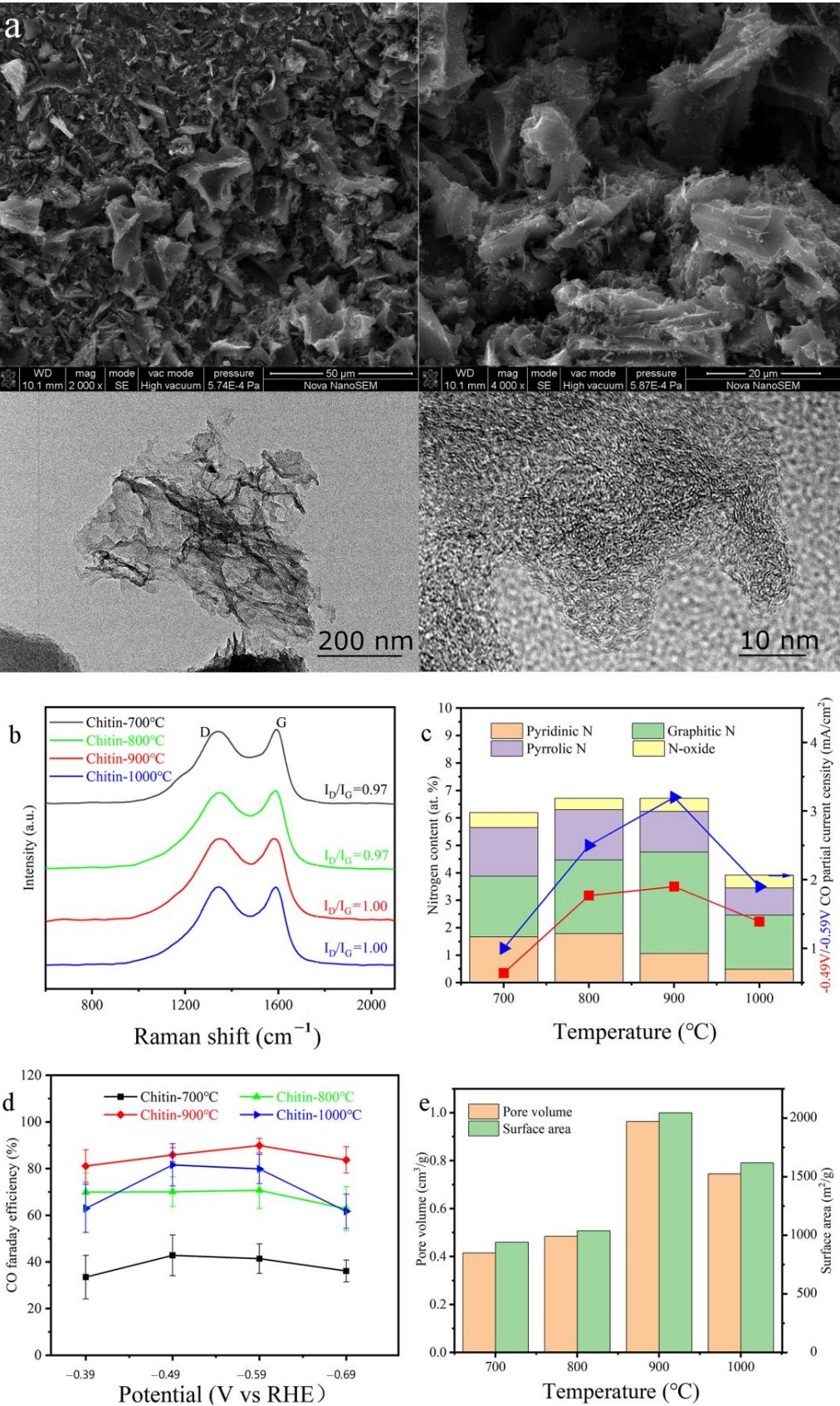

**Figure 5.** Characterization of chitin-derived catalysts prepared at different temperatures and their CO$_2$RR performance. (**a**). SEM and TEM at 1000 °C, (**b**). FE$_{CO}$, (**c**). Nitrogen contents and PCD at −0.59 V vs. RHE, (**d**). Raman, (**e**). Pore volume and surface area.

**Table 1.** Catalytic performance of reported metal-free catalysts for electrochemical reduction of $CO_2$ to CO.

| Catalyst | Specific Surface Area (m$^2$/g) | Electrolyte (M) | Potential (V vs. RHE) | $j_{CO}$ (mA/cm$^2$) | FE$_{CO}$ (%) | Ref |
|---|---|---|---|---|---|---|
| MWCNT/Cc | | 0.1 KHCO$_3$ | 0.56 | 0.27 | 88 | [49] |
| NCNTs-CAN-850 | | 0.1 KHCO$_3$ | 1.05 | 4 | 80 | [51] |
| N-CNT | | 0.1 KHCO$_3$ | 0.82 | 1.0 | 80 | [52] |
| NCNT-3-700 | | 0.5 KHCO$_3$ | 0.9 | 5.38 | ~90 | [53] |
| g-C$_3$N$_4$/MWCNT | 123.4 | 0.1 KHCO$_3$ | 0.64 | 0.55 | 60 | [50] |
| NS-C-900 | 160 | 0.1 KHCO$_3$ | 0.6 | 2.63 | 92 | [54] |
| NF-C-950 | 197 | 0.1 KHCO$_3$ | 0.6 | 1.9 | 90 | [55] |
| P-OLC | 338 | 0.5 NaHCO$_3$ | 0.9 | 4.9 | 81 | [56] |
| NPC-900 | 545 | 0.5 KHCO$_3$ | 0.67 | 2.3 | 95 | [57] |
| SaU-900 | 662 | 0.1 KHCO$_3$ | 0.85 | 2 | 22 | [58] |
| NRMC-900-3 | 832 | 0.1 KHCO$_3$ | 0.6 | 2.9 | 82 | [59] |
| NDC-700 | 1269 | 0.5 NaHCO$_3$ | 0.71 | 12.5 | 84 | [35] |
| S, N-carbon | 1332 | 0.1 KHCO$_3$ | 0.99 | 0.47 | 11.3 | [60] |
| N-BAX-M-950 | 1494 | 0.1 KHCO$_3$ | 0.66 | 0.7 | 40 | [34] |
| A-350-1000 | 1500 | 2 KHCO$_3$ | 1.1 | 1.5 | 89 | [37] |
| CB-NGC-2 | 1673.6 | 0.1 KHCO$_3$ | 0.56 | 3.7 | 91 | [36] |
| Chitin-900 °C | 1972 | 0.1 KHCO$_3$ | 0.59 | 3.3 | 90 | This work |

*2.5. Investigation of Carbon Electrocatalysts Mechanism for CO$_2$RR*

The Tafel plots were derived to explore electrodynamics for the CO formation (Figures 6a and S8). The chitin-900 °C exhibited the lowest Tafel slope, suggesting the fastest kinetics for CO formation. In addition, the overpotential of the chitin-900 °C was lower than the others at the same exchange current density (Figures 6a and S8), indicating that the chitin-900 °C was a better catalyst than the others. Since the Tafel slope is not close to 118 mV dec$^{-1}$, a value for the rate determining the step at which CO$_2$ obtains one electron to form the CO$_2^{\bullet-}$ key intermediate, the complexity of the reaction system is demonstrated. The EIS showed a single semi-circle in the frequency range of 10$^1$–10$^5$ Hz, revealing the single charge transfer process of CO$_2$RR (Figure 6b). The chitin-900 °C showed the lowest interfacial charge transfer resistance (R$_{ct}$), leading to rapid electron transfer in the CO$_2$ reduction process [61], which is in line with the Tafel results. The EIS results (the inset in Figure 6b) further showed the highest double layer capacitance (C$_{dl}$) for the chintin-900 °C, suggesting that it contained a more active surface, leading to the CO$_2$RR performance being enhanced.

Since CO and H$_2$ are the only products produced, the CO$_2$RR process is initiated by the adsorption of CO$_2$ when CO$_2$ approaches the surface, resulting in a partially negative and bound *CO$_2$ intermediate, followed by a CO$_2^{\bullet-}$ or *COOH intermediate, subsequently forming a *CO species and CO release after desorption from the catalyst surface. Hence, we examined the CO$_2$ adsorption of the prepared catalysts. It was found that the higher the pore volume and surface area of porous catalysts (<1 nm) were, the greater the CO$_2$ adsorption, showing that the CO$_2$ adsorption was correlated with ultra-narrow pores less than ≤1 nm in diameter (Figures 6c,d and S9). Despite the fact that pyridine nitrogen and pyrrolic nitrogen are thought to efficiently bind with CO$_2$ due to excess negative charges, no clear relationship between the CO$_2$ adsorption with their content (Figures 5c and 6e) was observed. All these results together indicated that the CO$_2$ adsorption predominantly depends upon ultra-narrow pores while the nitrogen-content plays a complementary role.

ΔH$_{ads}$ between the chitin-derived catalysts and CO$_2$ were further determined, all in the typical physisorption range of 25.0–34 kJ/mol (Figure 6f). The higher the nitrogen content, the higher the initial ΔH$_{ads}$, indicating that the initial ΔH$_{ads}$ was positively correlated with the nitrogen content, revealing that nitrogen functionalities enhanced interactions between CO$_2$ molecules and the catalyst surface, further leading to the improved CO$_2$ adsorption. With the CO$_2$ coverage elevation, the ΔH$_{ads}$ gradually decreased except for the chitin-

1000 °C (Figure 6f), implying less surface area occupied with nitrogen functional groups. In comparison with the chitin-700, the chitin-1000 °C contained less pyridinic and graphitic nitrogen but more $CO_2$ adsorption and higher $FE_{CO}$, implying that the ultra-narrow pore is beneficial to the $CO_2$ adsorption, which is further conducive to improving the electrical catalyst property.

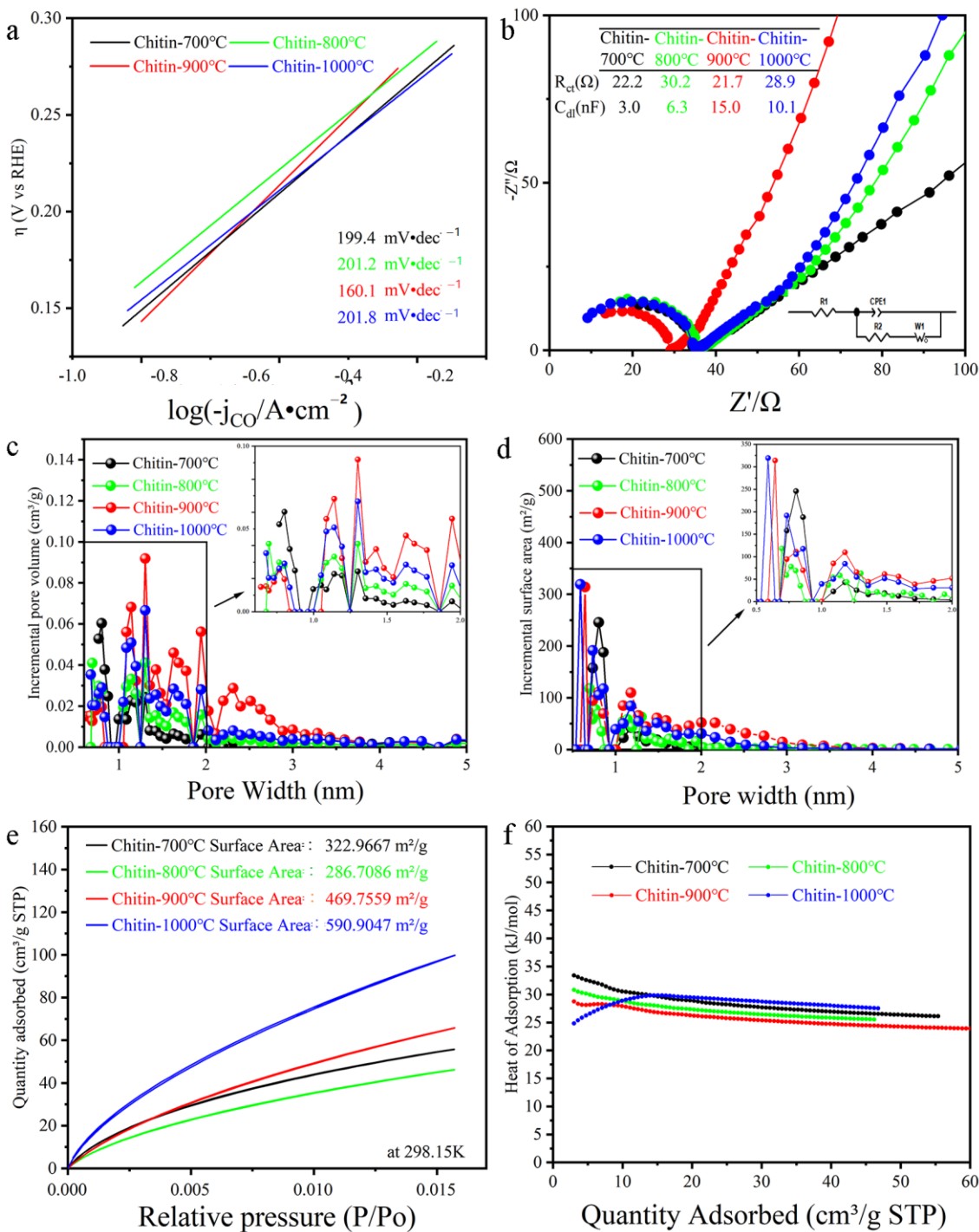

**Figure 6.** Tafel, EIS and $CO_2$ absorption heat of chitin-derived catalysts. (**a**). Tafel slope, (**b**). EIS analysis conducted at the potential of −0.59 V vs. RHE, (**c**). BET analysis of incremental pore volume and surface area for $N_2$ adsorption at 273 K, (**d**,**e**). $CO_2$ isotherm absorption at 25 and 0 °C, (**f**). $CO_2$ absorption heat.

Overall, the best $CO_2RR$ performance of the chitin-2.5Fe is associated with the synergetic effects of high pyridinic and graphitic nitrogen content for more active sites, more ultra narrow pores and meso-pores for facilitating $CO_2$ adsorption and mass transfer, high surface area for providing more active surface and low interfacial charge transfer resistance for rapid electron transfer.

## 3. Materials and Methods

### 3.1. Materials

Chitin (99%) was purchased from Shenzhen Lefu Tech. (Shenzhen, Guangdong, China). Potassium bicarbonate ($KHCO_3$, 99.5%) and $FeCl_3$ (99%) were purchased from Wuxi Yatai United Chemical (Wuxi, Jiangsu, China). $KHCO_3$ (99%) was purchased from Macklin (Shanghai, China). $ZnCl_2$ (98%) was from Shanghai Taitan Tech. (Shanghai, China). Hydrochloric acid (HCl, 36%–38%), chitosan (degree of deacetylation 90%) and anhydrous ethanol were from Sinopharm Chemical Reagent (Shanghai, China). Nafion 117 (5%) and Nafion 117 membrane were purchased from DuPont (Wilmington, DE, USA). $(NH_4)_2FeSO_4$ (98%) and $K_2FeO_4$ (98%) was purchased from Shanghai Aladdin Bio-Chem Technology (Shanghai, China). Carbon paper was purchased from Beijing JingLongTeTan Technology (Beijing, China). Unless otherwise mentioned, all chemicals were used as received without further purification. $N_2$ and $CO_2$ and Ar were obtained from Nanjing special gas (Nanjing, Jiangsu, China).

### 3.2. Biomass-Derived Electrocatalyst Preparation

Fine chitin and chitosan powders were mixed with an activator of anhydrous $FeCl_3$ at a certain weight ratio in a porcelain boat. Then, the boat with the mixtures was placed into a tube furnace for 1 h of nitrogen exchange. The samples were then heated at 5 °C/min to 400 °C under steady $N_2$ flow at 10 mL/min as a protective gas. After 1 h, the temperature continued to rise at a ramp rate of 5 °C/min. The catalysts were then prepared with the ratio of chitin to Fe at 1:2.5 at the different temperatures of 700, 800, 900 and 1000 °C for 1 h under an $N_2$ atmosphere denoted as chitin-Fe-700, chitin-Fe-800, chitin-Fe-900 and chitin-Fe-1000, respectively. Furthermore, the catalysts were prepared with the various ratios of chitin to Fe at 1:2, 1:2.5, 1:3 and 1:4 at 1 h 400 °C and 1 h 900 °C under $N_2$, denoted as chitin-2.0Fe, chitin-2.5Fe, chitin-3.5Fe and chitin-4.0Fe, respectively. For comparison, pure chitin-derived catalyst (chitin-0Fe) was also prepared in the absence of $FeCl_3$. After activation and carbonization, the cooled catalysts were immersed in excess 0.1 M HCl and stirred for 24 h, then washed with 0.1 M HCl several times to remove all residues containing Fe [36,37], then thoroughly washed with deionized water until the pH reached neutral and then washed with ethanol. The catalysts were collected after filtration and dried in a vacuum oven at 60 °C overnight.

### 3.3. Electroreduction of $CO_2$ and Electrochemical Characterizations

Electrochemical $CO_2$ reduction reaction was carried out using a H-cell separated by Nafion 117 membrane with three electrodes on an electrochemical workstation (CHI 604e, Shanghai Zhenhua, Shanghai, China). A platinum foil (Jingke Instrument, Shanghai, China), an electrocatalyst-coated carbon plate with dimensions of 1 cm × 2 cm and a saturated calomel electrode (SCE) (Shanghai Chuxi, Shanghai, China) were used as the working electrode, the counter electrode and the reference electrode, respectively. The working electrode was prepared by mixing 5 mg of carbonized activated biomass-based catalyst with 480 μL ethanol, 480 μL deionized water and 50 μL Nafion solution, ultrasonicated for 1 h until the mixture became uniform. The mixture was then coated onto a carbon plate in an area of 1 cm × 1 cm on each side and dried overnight. Prior to the electroreduction of $CO_2$, 30 mL of 1 M $KHCO_3$ was used as an anolyte while 1 M $KHCO_3$ saturated with $CO_2$ (pH 7.4) was used as a catholyte. $CO_2$ continued to flow into the catholyte at a rate of 30 mL/min during $CO_2RR$. All the experiments were performed at 20 °C, giving the SCE a

potential of +0.244 V. All the potentials performed in this study were referred to the RHE unless otherwise noted: E (vs. RHE) = E (vs. SCE) + 0.244 + 0.059 pH.

Cyclic voltammetry (CV) was carried out for 30 cycles from $-0.6$ to $-2.0$ V (vs. the $Hg/Hg_2Cl_2/KCl$ electrode) at a scan rate of 50 mV·s$^{-1}$. Linear sweep voltammetry (LSV) was carried out in a range between $-0.39$ and $-1.79$ V at a scan rate of 10 mV·s$^{-1}$. Electrochemical impedance spectroscopy (EIS) analysis was performed at $-0.91$ V vs. SHE using a frequency module of the potentiostat in a range of 10 Hz–10,000 Hz (at 10 mV AC amplitude). The obtained data were fitted with Corrtest software. The $CO_2RR$ products were analyzed using a gas chromatography (GC9790, Zhejiang Fuli Instrument, Wenling, Zhejiang, China) with a Porapak N column and a 5A Molecular sieve (Wuhan Puli, Wuhan, Hubei, China). Faradaic efficiency (*FE*) and partial current density (*PCD*) were determined using the following equations:

$$FE \ (\%) = \frac{znF}{I \times t} \times 100 \tag{1}$$

$$PCD = FE \ (Product) \times \frac{I}{A} \tag{2}$$

where $n$ is the number of moles of product, $I$ is the total current at the time of sample collection (mA), $z$ is the number of electrons, $F$ is the Faradaic constant (C mol$^{-1}$), $t$ is the time for sample collection from the electrochemical cell (s), *FE* is the Faradaic efficiency and $A$ is the electrode geometric area (cm$^2$).

### 3.4. Characterization

Catalyst morphology and microstructure were analyzed using FEI Nova nano SEM 430 (FEI, Hillsboro, OR, USA) and a transmission electron microscope (TEM) (JEM-2100F, JEOL, Tokyo, Japan). X-ray diffraction (XRD) was carried out to determine crystalline structures of catalysts by Rigaku SmartLab SE (Tokyo, Japan) with CuK$\alpha$ as a radiation source. X-ray photoelectron spectroscopy (XPS) (Thermofisher Nexsa, Thermo Fisher, Illkirch-Graffenstaden, France) was used to characterize surface compositions using AlK$\alpha$ as a radiation source. The specific surface area was measured using an ASAP 2460 (Micromeritics, Norcross, GA, USA) with $N_2$ adsorption/desorption at 77 K, and pore size distribution (PSD) was calculated by applying the NLDFT model. $CO_2$ adsorption was performed at 273 and 298 K. A laser Raman spectrometer was applied to obtain a Raman spectrum in a scan range of 500–4000 cm$^{-1}$ with a laser source of 532 nm (LabRAM HR Evolution, Horiba, Kyoto, Japan). The samples were observed using a nano SEM 430 at 50 eV–15 keV with an Everhart–Thornley Detector (ETD).

## 4. Conclusions

A nitrogen self-doped metal-free catalyst was prepared from inexpensive biomass-chitin via a simple one-step pyrolysis for electrochemical $CO_2RR$. The as-prepared electrocatalyst of chitin-900 with a surface area of 1972 m$^2$/g was found to convert $CO_2$ into CO with FE$_{CO}$ of ~90% and PCD of 3.3 mA·cm$^{-2}$ at a potential of $-0.59$ V (vs. RHE). This good $CO_2RR$ performance results from the synergetic effects of plentiful active sites, rich ultra-micropores beneficial to $CO_2$ adsorption, abundant mesopores for good $CO_2$ transport, high total content of pyridinic and graphitic nitrogen and a low interfacial charge transfer resistance. This study shows the feasibility of an N self-doped biomass-derived catalyst for $CO_2RR$ with the potential for large-scale industrial applications.

**Supplementary Materials:** The following supporting information can be downloaded at: https://www.mdpi.com/article/10.3390/catal13050904/s1, Figure S1: TGA of chitin-derived catalysts prepared at different ratios of Chitin to $FeCl_3$, Figure S2: Oxygen content of chitin-derived catalysts prepared at different ratios of Chitin to $FeCl_3$, Figure S3: Carbon content of chitin-derived catalysts prepared at different ratios of Chitin to $FeCl_3$, Figure S4: Cumulative pore volume of chitin-derived catalysts prepared at different ratios of Chitin to $FeCl_3$, Figure S5: Morphology of chitin-derived

catalysts prepared at different temperatures of 700–1000 °C, Figure S6: XPS analysis of chitin-derived catalysts prepared at different temperatures of 700–1000 °C, Figure S7: FE of catalysts. top: $FE_{H2}$ of chitin-derived catalysts prepared at different temperatures of 700–1000 °C. bottom: $FE_{CO}$ and $FE_{H2}$ of a carbon plate without coating, Figure S8: LSV of chitin-derived catalysts prepared at different temperatures of 700–1000 °C, Figure S9: BET analysis of incremental pore volume and surface area for $N_2$ adsorption at 273 K.

**Author Contributions:** Conceptualization, X.X.; methodology, P.S. and X.W.; validation, P.S. and N.A.; formal analysis, X.X. and M.Z.; investigation, P.S.; resources, K.Z.; data curation, X.W.; writing—original draft preparation, X.X.; writing—review and editing, X.X.; visualization, M.Z.; supervision, X.X.; project administration, X.X. and K.Z.; funding acquisition, X.X. All authors have read and agreed to the published version of the manuscript.

**Funding:** This work was supported by the National Natural Science Foundation of China (91534107 and 21978001), the leader in Academic and Technology in Anhui Provence (2021D306) and the Start Fund for Biochemical Engineering Research Centre from Anhui University of Technology.

**Data Availability Statement:** Data is contained within the article or Supplementary Materials.

**Conflicts of Interest:** The authors declare no conflict of interest.

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
