# Peer review of "Nitrogen Self-Doped Metal Free Catalysts Derived from Chitin via One Step Method for Efficient Electrocatalytic CO2 Reduction to CO"

_catalysts, doi:10.3390/catal13050904_

Round 1

Reviewer 1 Report

This manuscript by Xu et al. presented the carbonization of low-cost biomass chitin to prepare carbon catalysts for CO2RR. The development of metal-free electrocatalysts is always being pursued as promising alternatives to noble metals. The results from this work will thus be interested to researchers in this community. The introduction part is well put and the experimental detailed are given. Systematic characterizations are done to support the derived conclusions. However, before I can recommend the publication of this work, I would like to discuss several fundamental issues as noted below: 1.      In the title, “metal-free” is used, but FeCl3 is used in chitin pyrolysis. How can the authors be sure that the produced carbon is Fe-free. From the TEM images in Figure 2, some dark particles that could be iron species are clearly seen. 2.      The identification of active N configuration is accomplished following a questionable principle, which is the most abundant species in the best-performing catalysts are the most active. Accordingly, the authors reckon that graphitic N is CO2RR-active. However, graphitic N has better thermal stability than pyridinic and pyrrolic N, so it comes naturally that graphitic N content will increase as calcination temperature increases. The aforesaid conclusion is thus not reasonable. 3.      The font type, size in the Figures should be unified and clearly presented. 4.      Regarding the defect engineering, some recent references are suggested (Chin. J. Inorg. Chem. 2022, 38(11), 2113-2126; Green Energy Environ. 2022, DOI: 10.1016/j.gee.2022.11.002)

5.      There should NOT be any significant figures in the BET specific surface areas.

The English language should be polished.

Author Response

This manuscript by Xu et al. presented the carbonization of low-cost biomass chitin to prepare carbon catalysts for CO2RR. The development of metal-free electrocatalysts is always being pursued as promising alternatives to noble metals. The results from this work will thus be interested to researchers in this community. The introduction part is well put and the experimental detailed are given. Systematic characterizations are done to support the derived conclusions. However, before I can recommend the publication of this work, I would like to discuss several fundamental issues as noted below:

  1. In the title, “metal-free” is used, but FeCl3is used in chitin pyrolysis. How can the authors be sure that the produced carbon is Fe-free. From the TEM images in Figure 2, some dark particles that could be iron species are clearly seen. 

In this paper, we prepared the chitin-based catalysts with different amount of FeCl3. To obtain biomass-derived metal free catalysts, the HCl washing method was used according to the previous studies [1, 2]. Furthermore, XPS and XRD were carried out. No peak associated with Fe element was found in XPS (Figure 1b) and no sharp peak corresponding to iron (1 1 0) at 44.9 o [3] was observed (Figure 1c), indicating that no Fe element in the catalysts was detected (Fig 1d and Fig 1c). As shown in Figure 2, dark aggregate was also observed for the catalysts prepared without using FeCl3, implicating that the dark particle could be some deposition. According to the XPS, XRD and TEM results, and methods used for preparing metal free biomass-based catalysts using HCl wash, the catalysts we prepared here was considered to be chitin-based metal free catalyst.

[1] X Hao, X An, AM Patil, P Wang, X Ma, X Du, X Hao, A Abudula, G Guan. Biomass-derived N‑doped carbon for efficient elec-trocatalytic CO2 reduction to CO and Zn−CO2 batteries. ACS Appl. Mater. Interfaces 2021. 13: 3738−3747. https://doi.org/10.1021/acsami.0c13440.

[2] M Chen, S Wang, H Zhang, P Zhang, Zi Tian, M Lu, Xi Xie, L Huang, W Huang. Intrinsic defects in biomass-derived carbons facilitate electroreduction of CO2. Nano Res. 2020. 13: 729–735. https://doi.org/10.1007/s12274-020-2683-2.

[3] Pure iron nanoparticles prepared by electric arc discharge method in ethylene glycol. The European Physical Journal Applied Physics 59(3):30401. DOI:10.1051/epjap/2012110303

We added the references in the methods, line 112, and discussion in line 202, 206-207, 231-236.

  1. The identification of active N configuration is accomplished following a questionable principle, which is the most abundant species in the best-performing catalysts are the most active. Accordingly, the authors reckon that graphitic N is CO2RR-active. However, graphitic N has better thermal stability than pyridinic and pyrrolic N, so it comes naturally that graphitic N content will increase as calcination temperature increases. The aforesaid conclusion is thus not reasonable.

Point taken. We rephrased the statement, line 293-295, line 327-329.

  1. The font type, size in the Figures should be unified and clearly presented.

Done.

  1. Regarding the defect engineering, some recent references are suggested (Chin. J. Inorg. Chem. 2022, 38(11), 2113-2126; Green Energy Environ. 2022, DOI: 10.1016/j.gee.2022.11.002).

Done.

  1. There should NOT be any significant figures in the BET specific surface areas.

No significant figures in the BET specific surface areas.

Reviewer 2 Report

Experiments and analysis performed by the authors are pretty detailed in the study. I do have some comments on the study and also regarding the motivation.

1. I agree that biomass based solutions are encouraging from a sustainable source point of view. But the challenge of abundance remains. That needs to be addressed in the study. Chitin obtained from crabs or shrimps - how sustainable is that, in terms of production volume when compared to the amount of CO2 that needs to be removed or converted to CO? This aspect needs to be mentioned and brought into perspective for the reader.

2. I would encourage mentioning the composition of chitin used in the study, and also how it varies based on the source.

3. There are some errors in molecular formulae, especially F written instead of Fe. Please rectify them.

4. FE is mentioned early on in the study with no definition. Would be nice to define abbreviations when first used.

5. The figures are not clear and is hard to read the data on them. They seem tightly placed for the reader. Please provide good resolution large images for ease of reading.

6. Chitin-Fe2.5 is an interesting material which shows performance different from the trend. But the explanation as to why this one is different or better is weak or missing. I do not seem to locate such a write up in the article. Would recommend strengthening this section since this seems to be one of the main points of the article.

7. If I get this right, Fe is used as a catalyst on Chitin supports. The reason Chitin is used is mainly due to the biomass sources, cheap, and ability to have a large surface area and pore volume. How different or novel is this work compared to previous work on Chitin supports? Is it the synthesis procedure? This has not come out clearly on the article. Would recommend highlighting that throughout the text since this is the central idea of the manuscript.

8. On the Raman spectra, Fig 1b, D/G is not significantly different for the different catalysts (0.84 vs 1.0). So not sure of the conclusions derived from this.

9. It was good to see the comparison in Table 1. Would be nice to have it early on in the study to give the reader an idea of the existing catalyst space.

10. Additionally, to introduce large scale use of metal free catalysts, comparing them with state of the art metal catalysts is crucial. It would be good to include them as the reference as well. This will help present the overall CO2RR space. And also guide the scientists and readers towards further developing new catalysts.

10. How stable is the proposed Chitin based materials? How long were the runs? Stability is a major issue with electrocatalysts. Would recommend talking about this aspect in the text.

11. How does Chitin derived from different sources affect the generalization of this catalyst synthesis?

Overall, the article is detailed and presents Chitin as a good support for the electrocatalysts. It also presents an option of using Chitin for other catalysts too, where high SA or pore volume is desired.

Looks good. No major comments. Minor ones have been mentioned in the comments above.

Author Response

Experiments and analysis performed by the authors are pretty detailed in the study. I do have some comments on the study and also regarding the motivation.

  1. I agree that biomass based solutions are encouraging from a sustainable source point of view. But the challenge of abundance remains. That needs to be addressed in the study. Chitin obtained from crabs or shrimps - how sustainable is that, in terms of production volume when compared to the amount of CO2 that needs to be removed or converted to CO? This aspect needs to be mentioned and brought into perspective for the reader.

         We rewrote this part in the introduction, line 67-72.

  1. I would encourage mentioning the composition of chitin used in the study, and also how it varies based on the source.

Good point. The composition of chitin could be a factor for affecting the nitrogen content in the catalysts, which is going be investigated in the future study.

  1. There are some errors in molecular formulae, especially F written instead of Fe. Please rectify them.

Done.

  1. FE is mentioned early on in the study with no definition. Would be nice to define abbreviations when first used.

The definition was given in line 141.

  1. The figures are not clear and is hard to read the data on them. They seem tightly placed for the reader. Please provide good resolution large images for ease of reading.

Replaced the Figures with high resolution.

  1. Chitin-Fe2.5 is an interesting material which shows performance different from the trend. But the explanation as to why this one is different or better is weak or missing. I do not seem to locate such a write up in the article. Would recommend strengthening this section since this seems to be one of the main points of the article.

The properties including morphology, FECO, jCO, N type and content, surface are, pore volume, CO2 adsorption and EIS were determined for the catalysts prepared at different conditions. The chitin-2.5Fe exhibited the best CO2RR performance because of the synergetic effects of high pyridinic and graphitic nitrogen content for more active sites, more ultra narrow pores and meso-pores for facilitating CO2 adsorption and mass transfer, high surface area for providing more active surface, low interfacial charge transfer resistance for rapid electron transfer.

We added some discussion, line 394-398.

  1. If I get this right, Fe is used as a catalyst on Chitin supports. The reason Chitin is used is mainly due to the biomass sources, cheap, and ability to have a large surface area and pore volume. How different or novel is this work compared to previous work on Chitin supports? Is it the synthesis procedure? This has not come out clearly on the article. Would recommend highlighting that throughout the text since this is the central idea of the manuscript.

 In this study we prepared the chitin-derived metal free catalysts using a simple one-step pyrolysis procedure instead of more complicated procedures in the previous studies. To highlight this, we emphasize this in the abstract and conclusion.

  1. On the Raman spectra, Fig 1b, D/G is not significantly different for the different catalysts (0.84 vs 1.0). So not sure of the conclusions derived from this.

That is right. In line 199-200, we wrote “A slightly higher ID/IG value suggested more intrinsic defects in chitin-2.5Fe probably including lattice vacancies and edge dislocations.”

  1. It was good to see the comparison in Table 1. Would be nice to have it early on in the study to give the reader an idea of the existing catalyst space.

That is a good point. In this study, we aimed to compare our work with the carbon-based catalysts not just biomass-derived catalysts. Hence at the end of this paper we put all these data together in Table 1 to let readers know how good the CO2RR performance of the chitin-derived catalysts prepared using the one-step pyrolysis procedure.        

  1. Additionally, to introduce large scale use of metal free catalysts, comparing them with state of the art metal catalysts is crucial. It would be good to include them as the reference as well. This will help present the overall CO2RR space. And also guide the scientists and readers towards further developing new catalysts.

Good point. Currently large scale preparation of metal free catalysts is critical for their future large scale applications in CO2RR. In this study, a simple method was provided for preparing metal-free catalysts using chitin as precursors, and their CO2RR performance was further determined. The large scale use of the chitin-derived catalysts is needed to be discussed in the further study.

  1. How stable is the proposed Chitin based materials? How long were the runs? Stability is a major issue with electrocatalysts. Would recommend talking about this aspect in the text.

Good point. We run the stability test for the chitin-derived catalysts which maintained the stable CO2RR performance for around 13 h but not present in this study. In the future study, we will further improve the catalyst property and discuss the stability in detail.

Fig Stability of the chitin-derived catalyst at -0.59 V (vs RHE)

  1. How does Chitin derived from different sources affect the generalization of this catalyst synthesis?

Good point. Currently we just looked into the effect of preparation conditions on the catalyst property. The effect of chitin source is needed to be investigated in the future study.

Reviewer 3 Report

The chitin-derived electrocatalysts are interesting for CO2RR and the whole experimental presentation is complete.  Authors can add an additional case of the CO2RR performance with a carbon plate electrode without coating as a comparison. 

Minor edition of English language required.

Author Response

The chitin-derived electrocatalysts are interesting for CO2RR and the whole experimental presentation is complete.  Authors can add an additional case of the CO2RR performance with a carbon plate electrode without coating as a comparison. 

Done. The added Figure (Fig S7) showed very low FECO by a carbon plate without coating, indicating.  We added the discussion in line 283-285.

Round 2

Reviewer 1 Report

I am happy to recommend the acceptance of this work now.

Pretty good.